# Comparative Seasonal Respiratory Virus Epidemic Timing in Utah

**DOI:** 10.3390/v12030275

**Published:** 2020-02-29

**Authors:** Zayne Y. Callahan, Trevor K. Smith, Celeste Ingersoll, Rebecca Gardner, E. Kent Korgenski, Chantel D. Sloan

**Affiliations:** 1Department of Public Health, Brigham Young University, Provo, UT 84602, USA; zayneycallahan@gmail.com (Z.Y.C.); trevorkent1@gmail.com (T.K.S.); 2Department of Statistics, Brigham Young University, Provo, UT 84602, USA; celesteingersoll3@gmail.com (C.I.); rbecca22@gmail.com (R.G.); 3Pediatric Clinical Program, Intermountain Health Care, Salt Lake City, UT 84102, USA; Kent.Korgenski@imail.org

**Keywords:** influenza, viral interference, wavelet

## Abstract

Previous studies have found evidence of viral interference between seasonal respiratory viruses. Using laboratory-confirmed data from a Utah-based healthcare provider, Intermountain Health Care, we analyzed the time-specific patterns of respiratory syncytial virus (RSV), influenza A, influenza B, human metapneumovirus, rhinovirus, and enterovirus circulation from 2004 to 2018, using descriptive methods and wavelet analysis (*n* = 89,462) on a local level. The results showed that RSV virus dynamics in Utah were the most consistent of any of the viruses studied, and that the other seasonal viruses were generally in synchrony with RSV, except for enterovirus (which mostly occurs late summer to early fall) and influenza A and B during pandemic years.

## 1. Introduction

Seasonal viruses are responsible for hundreds of thousands of deaths and extensive morbidity in temperate climates each year [1,2]. In the United States, epidemics of respiratory syncytial virus (RSV) and influenza typically begin in the Southeast United States and progress to the Northwest, through the months of October to April [3,4,5]. Much of the seasonal timing and geographic spread between respiratory viruses coincide, resulting in high prevalence of coinfection globally, which may potentially be linked to disease severity [6,7,8].

Competition for host cells during coinfection can result in viral interference in the form of delaying or preventing infection by the secondary virus [9,10]. It is possible that this cellular interference may be detectable on a population level. For example, research suggests that epidemics of RSV, coronavirus, and influenza B can be respectively delayed, intensified, or inhibited if circulation of influenza A begins early (before week one of a given seasonal year) [11]. Furthermore, faster growing seasonal viruses, such as rhinovirus, may reduce the rate of replication of slower growing seasonal viruses, while RSV infection may in turn reduce the risk of rhinovirus coinfection [9,12].

The complex ecological interactions between viruses are difficult to discern using typical measures of correlation, since relationships between epidemics may shift over time, depending on strain severity, host immunity, and climate factors. In this study, we compared epidemic timing and calculated phase differences between seasonal epidemics of RSV, influenza A and B, metapneumovirus, enterovirus, and rhinovirus, using descriptive and wavelet analyses. Wavelet analysis allows for viewing changes in epidemic synchronistic patterns over multiple years [13,14].

## 2. Materials and Methods

### 2.1. GermWatch^®^ Data

We obtained weekly frequency data for six seasonal respiratory viruses (influenza A and B, RSV, metapneumovirus, enterovirus, and rhinovirus), from Intermountain Healthcare’s GermWatch^®^ database, for the 2005–2006 winter season to the 2017–2018 winter season. We did not have data for summer 2018, nor did we have access to data at the individual level. Intermountain Healthcare is the largest health-care provider in the Intermountain West region of the United States and operates a nonprofit system of 22 hospitals and more than 190 clinics. The study was considered exempt by the Institutional Review Board at Brigham Young University.

The principal laboratory methods of virus identification were direct fluorescent antibody (DFA), viral culture, and rapid antigen testing between 2000 and 2007. After 2007, polymerase chain reaction (PCR) methodology became the primary method of identification, but DFA and rapid antigen testing was still available. Due to the aggregated nature of our dataset, it is not possible to differentiate between which tests were used for each data point. However, the more sensitive method (i.e., PCR > DFA > rapid antigen testing) was used in the analysis if different methods were ordered and to exclude duplicate test results.

### 2.2. Analysis

We conducted two separate analyses to compare epidemic timing between viruses over the study period. The first was a descriptive analysis in which epidemic initiation, peak, and termination were identified based on a change point model. Second, we conducted a wavelet analysis to compare epidemic synchrony across seasons. All analyses were conducted in R (v. 3.5.1) [15]. Note that five of the viruses studied occur during the fall and winter seasons, while enterovirus infection typically occurs during late summer in temperate climates.

Wavelet analysis and change point models are complementary methods we implemented to investigate the timing and patterns of epidemics. For our purposes, the strength of change point models over others, such as SIR models or circular statistics, is that they give more precise estimates of when different phases of a past epidemic began and terminated. Wavelet analysis compares epidemic timing in a way that allows for the behavior of the epidemic to change in different years. This would be more difficult if using methods that employ sine waves to estimate epidemic curves. We can therefore make direct comparisons of the historical characteristics of the epidemics (via change point models) and their synchrony with one another (wavelet analysis), using our selected methods. We recognize that there are many other methods that would lend different insights into the epidemic patterns described, but we deemed these two sufficient for the current study.

#### 2.2.1. Descriptive Analysis

For the descriptive analysis, we used a statistical change point model that was recently utilized to analyze RSV circulation throughout the United States [16]. We used this model to determine seasonal patterns in epidemic initiation, peak, and termination for all six viruses previously mentioned. For each year and virus, a base value count was determined as a starting value for the curve. Afterward, slopes of the mean counts were found, starting at base value, and four change points were identified: T1–T4. These data were used to create a line plot depicting the mean frequency count for each virus during each seasonal year.

A seasonal year is defined as starting on 1 July and ending on 30 June of the following year (e.g., seasonal year 2004 represents 1 July 2003 to 30 June 2004). The line between change points T1 and T2 represents the slope of virus onset to epidemic peak. The line between change points T2 and T3 represents the epidemic peak duration. The line between change points T3 and T4 represents the virus decline to termination (Figure 1).

Data are lacking for metapneumovirus prior to January 2007 and for rhinovirus prior to November 2007. Enterovirus is shown in a separate line plot due to its having a significantly lower mean count than the five other viruses, and it is plotted on a 1 January to 31 December cycle rather than the regular seasonal year.

In order to validate our method, we compared our estimates of peak and epidemic duration of RSV with those reported by the National Respiratory and Enteric Virus Surveillance System (NREVSS) [17]. Denver was the closest NREVSS surveillance site and is generally used to describe trends in the Western Region of the United States. NREVSS indicates onset, peak, and offset dates for RSV trends. Of the 9 seasonal years that NREVSS indicated peak dates, 7 of them fell within our peak durations (T2 and T3).

#### 2.2.2. Wavelet Analysis

To conduct the wavelet analysis, we first normalized the frequency data to have a mean of zero. We then calculated the restructured component of the epidemic curves, using a period window of 32 to 65 weeks for each virus, except for influenza A and B. For the influenza viruses, a longer period of 108 weeks (2 years) was necessary to accurately capture epidemic patterns in pandemic years (2009–2010). A loess smoother was applied to the rhinovirus data, to adjust for increases in laboratory testing in late 2007 and early 2008. The period windows were chosen based on cross-wavelet power levels with RSV (see Figure A4). RSV was selected as the reference due to its very consistent repeating annual pattern.

We plotted the phase angles and calculated the phase difference between each virus. Plots of each step of the analysis are found in Figure A3a–e. In Figure A3e, synchrony between the epidemic curves is determined by the phase difference being zero, while deviation from the zero line indicates the epidemics are out of phase. Wavelet analyses were done by using the R package WaveletComp [18].

Wavelet analysis is nuanced but has many features that are similar to other wave decomposition methods. It has been shown to be a useful tool in comparing infectious disease rates in several studies [13,19]. Its strength is in allowing the timing of the epidemic curves to change from year to year by using a wavelet instead of something more regular such as a sine curve. This allowed for greater flexibility and precision in identifying epidemic trends that may change from year to year.

## 3. Results

### 3.1. Descriptive Analysis

The dataset contained 28,671 laboratory-confirmed cases of RSV; 18,451 cases of influenza A; 4767 cases of influenza B; 1347 cases of Enterovirus; 4945 cases of metapneumovirus; and 31,281 cases of rhinovirus over 13 years (total = 89,462). (See Figure 2 for raw frequency plot of all viruses. See Table A1 for frequency data of each virus in each seasonal year.)

The highest and lowest frequencies of RSV were in seasonal year 2013 (4002 cases) and 2005 (506 cases), respectively. The majority of RSV cases occurred between January and March, with the most common month being February (see Table A2 for all monthly frequency data). Metapneumovirus rates were highest in seasonal year 2014 (821 cases) and lowest in 2008 (81 cases), with the majority of peak timing usually occurring between January and March, and the most common month being February.

The highest and lowest frequencies of influenza A were in seasonal year 2010 (3693 cases) and 2005 (132 cases), respectively. The earliest and latest timing of the peak of influenza A occurred in 2010 and 2012, respectively. Peak timing for influenza A was less consistent compared to other viral infections, primarily due to 2009 H1N1, but usually occurred between December and January.

In the change point analysis, influenza A exhibited the highest mean count in 2009 from 11 October to 18 October, with a peak mean count between 800 and 900, representative of the pandemic that year (Figure 3a). Influenza B exhibited its highest mean count in 2016, from 7 February to 28 February, with a peak mean count just above 150 (Figure 3b). RSV exhibited its highest mean count from 30 December 2012 to 17 February 2013, with a peak mean count between 307 and 362 (Figure 3c). Peak timing for influenza A overlapped eight of the 13 years with influenza B, and nine of the 13 years with RSV. (See Table A3 for all virus change point values and dates. See Figure A1 and Figure A2 for all virus change point plots.)

Enterovirus showed varied timing of peak mean frequency count and exhibited its highest mean count in 2010, from 8 October to 12 September, with a peak mean count just below 12 (Figure 3d). Metapneumovirus exhibited its highest mean count in 2010, from 14 February to 28 February, with a peak mean count between 95 and 100 (Figure 3a). Rhinovirus exhibited its highest mean count from 2 September 2012 to 12 May 2013, with a peak mean count of 111 (Figure 3c). Rhinovirus also showed the longest average yearly peak duration at a length of 27 weeks. RSV and influenza A showed the shortest average yearly peak durations at a length of five weeks (see Table A4 for all virus epidemic and peak durations).

### 3.2. Wavelet Analysis

Results of the wavelet analysis are shown in Figure A3, and cross-wavelet power spectrums are shown in Figure A4. Enterovirus and RSV show the least synchrony, as expected, with phase differences between approximately −50° and −180°. Influenza A and RSV show close synchrony with phase angles near 0°, except for a notable difference during the 2009–2010 H1N1 pandemic, and a shift in the 2013–2014 season. Influenza B was in synchrony with RSV, except for the 2008–2009 and 2013–2014 seasons. Metapneumovirus was out of phase with RSV in the 2011–2012 and 2013–2014 seasons, while rhinovirus moved in and out of synchrony with RSV over the study period.

## 4. Discussion

The strengths of the study include the use of a large, multiyear database of laboratory-confirmed samples and two complementary analysis methods, to determine synchrony in epidemic dynamics between six common respiratory viruses in Utah. However, interpretation of results is somewhat limited by the use of aggregate weekly frequency data rather than rates.

The first observation made from our data was visualizing the known and remarkable consistency with which RSV epidemics transmitted in Utah. The timing and duration of annual RSV epidemics were very similar across years, with peaks overlapping most years. Even in influenza pandemic years, RSV timing was not dramatically shifted.

Influenza B and metapneumovirus almost always had overlapping peaks and showed very similar trends in synchrony with RSV and overall timing. Both viruses typically peaked in February or March, but shifted to autumn during the 2010–2011 season. The wavelet analysis confirmed this phase shift, as well as showing a shift out of phase by enterovirus. The uneven pattern in enterovirus corresponds to shifts in peak timing from September to July, and then back to September. Pandemic influenza years (2009–2010) and seasons with high levels of H1N1 (2013–2014) coincided with more volatility in enterovirus and metapneumovirus synchronization.

Our results must be interpreted in light of the complex interactions that exist between viruses, populations, and the environment. Influenza A and B showed the most dynamic variability, while RSV remained markedly consistent in this population. Influenza is zoonotic, constantly experiencing genetic shift and drift, and spilling over into human populations. However, humans are the reservoir for RSV, and RSV is therefore very well adapted to human populations and may not be as driven by geographic and population-level factors as influenza. For example, temperature and humidity patterns are known to influence the spread of both RSV and influenza [20,21,22]. However, influenza epidemics are also influenced by sociodemographic characteristics of the underlying population [23,24]. It is unclear exactly how much RSV is dependent on similar population features, but previous studies support associations between respiratory viruses and human behaviors that may help explain their seasonality [25]. Previous studies showed that analysis of climate forcing and genetic drift and shift could well-predict influenza epidemic behavior [26]. Our study supports this conclusion, while also suggesting that influenza pandemic years are associated with irregularities in seasonal timing for certain other respiratory viruses. More research is needed to understand if this is due to interference by emerging influenza strains or due to some shared climate or population effect.

Rhinovirus is also well-adapted to humans but shows far less regularity in its timing than RSV. In the Utah data, rhinovirus persists at comparatively low peak levels over a much longer period of time. This is in line with other studies, some of which suggest rhinovirus may be well adapted to being present in populations at times other infections are not [27].

The delay of influenza season due to rhinovirus has been a subject of epidemiological interest for some time. For example, a particularly high-burden rhinovirus epidemic is hypothesized to have delayed the 2009 H1N1 pandemic in France [28]. Detecting interference at the population level may only be possible during more extreme events, such as pandemic influenza or higher-than-normal rhinovirus burden. Further studies with individual-level data and in different populations should be pursued to better understand why viral interference at the individual level does not necessarily translate to interference at the population level.

## Figures and Tables

**Figure 1 viruses-12-00275-f001:**
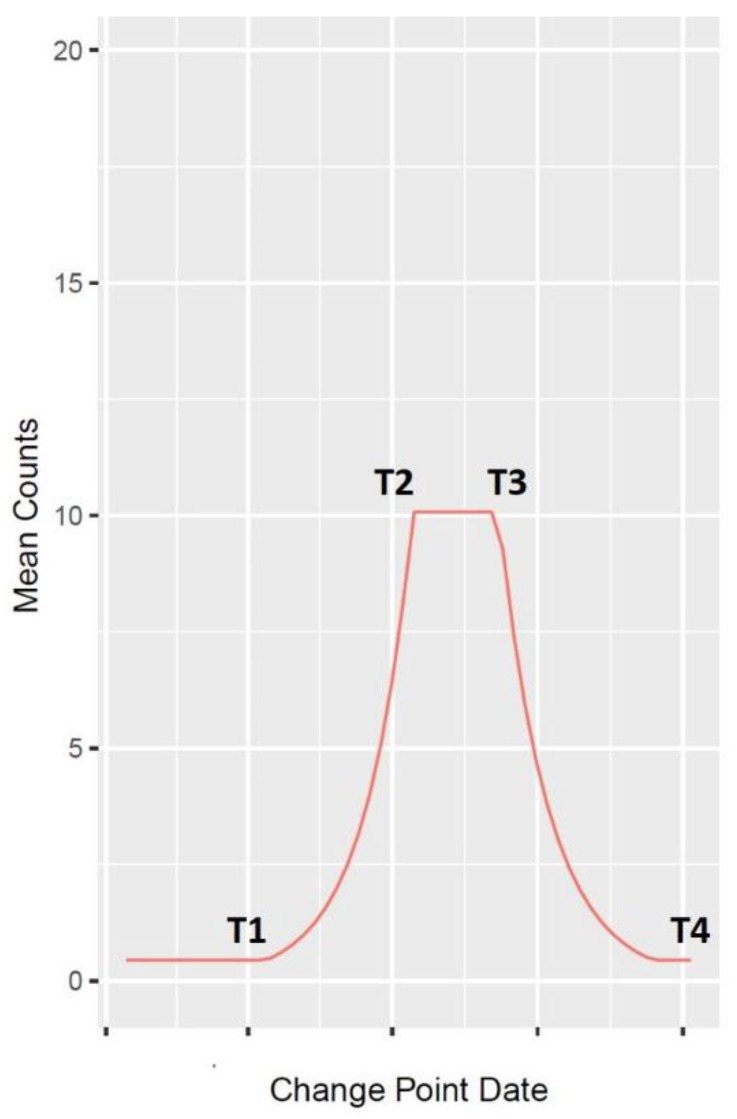
Change point plot description.

**Figure 2 viruses-12-00275-f002:**
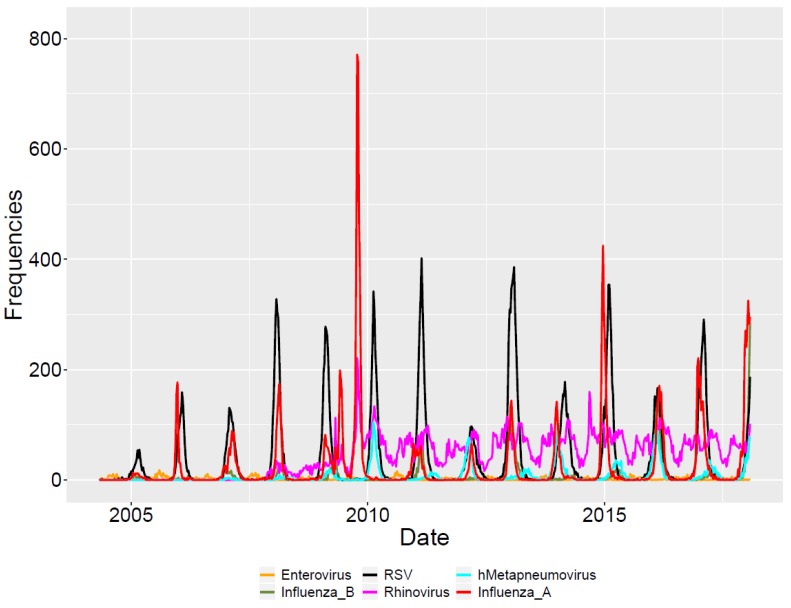
Raw frequency plot.

**Figure 3 viruses-12-00275-f003:**
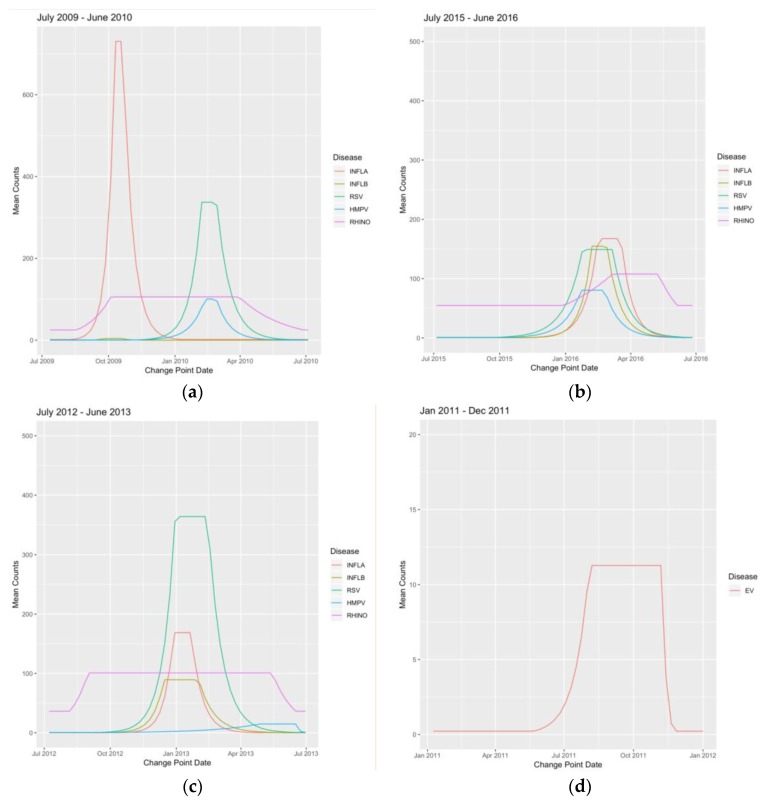
Change point plots: (**a**) influenza A and human Metapneumovirus exhibited their highest mean counts in the 2009–2010 season; (**b**) influenza B exhibited its highest mean count in the 2012–2013 season; (**c**) respiratory syncytial virus (RSV) and rhinovirus exhibited their highest mean counts in the 2012–2013 season; and (**d**) enterovirus (EV) exhibited its highest mean count in 2011.

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
