# Peer review of "Comparative Seasonal Respiratory Virus Epidemic Timing in Utah"

_viruses, 2020, doi:10.3390/v12030275_

Round 1

Reviewer 1 Report

The paper is well written with a fluent English. 

I have added a couple of stick notes to the text: one each  to the ethods  and discussion sections.

Most important, in the discussion I invite the authors to discuss the suitability of using Wavelet analysis to a relative simple seasonal event, and the advantage over traditional epidemiological methods to identify patterns of the event  otherwise unidentified. 

Author Response

Dear Reviewer, 

Thank you for your feedback on our paper. We feel it helped to improve the presentation of the study. We respond to each of your comments below:

1. I suggest to add a sentence to explain why the dataset was aggregated. Was a case definition be used for each virus isolate?

Response: Thank you. We are referring to the nature of the dataset we received, which was de-identified with frequencies by week, rather than individual cases. We clarified a sentence in the first paragraph of section 2.1 to reflect this:

"We did not have data for summer 2018, nor did we have access to any data at the individual level.”

2. I suggest to add a couple of sentences on the suitability of using Wavelet analysis to analyze epidemiological data and additional information revealed by this model over the traditional ones (i.e epidemic curves using rates etc etc)

Response: Thank you. We agree and added the below sentences:

"Wavelet analysis is nuanced but has many features that are similar to other wave decomposition methods. It has been shown to be a useful tool in comparing infectious disease rates in several studies (Viboud et al. 2006, Almeida, Codeco and Luz, 2018). Its strength is in allowing the timing of the epidemic curves to change from year to year by using a wavelet instead of something more regular such as a sine curve. This allowed for greater flexibility and precision in identifying epidemic trends that may change from year to year." 

Reviewer 2 Report

The manuscript by Callahan and colleagues describes the epidemic pattern of several important respiratory viruses, and attempts to determine if the time of onset, peak and offset of epidemics of each virus affect those of another virus. No clear pattern emerged, and the authors say the data are "complex" in the Discussion, without much in the way of hypotheses to explain the complexity.

A limitation of this paper is that many others have examined the temporal relationship of viral epidemics, and these are not referenced here. Specifically, the interaction of influenza and RSV epidemics has been studied several times, with no clear pattern identified.

Another limitation is a description of why the statistical methods used here are preferable to other methods. Non-epidemiologists especially would need to know this, as the methods are not widely understood

Perhaps the major limitation is the absence of a discussion of the meaning of the results. Where does this study advance our knowledge, and how do these data help us?

Author Response

Dear Reviewer, 

Thank you for your thoughtful review of the manuscript. We made several changes based on your feedback, and feel the paper is more clear and offers more insight. We respond to each of your comments below: 

1.) The manuscript by Callahan and colleagues describes the epidemic pattern of several important respiratory viruses, and attempts to determine if the time of onset, peak and offset of epidemics of each virus affect those of another virus. No clear pattern emerged, and the authors say the data are "complex" in the Discussion, without much in the way of hypotheses to explain the complexity.

Response: Thank you. We agree that the discussion was too short and did not offer enough insight into the results. We have expanded the discussion in section 4 as shown below:

"Our results must be interpreted in light of the complex interactions that exist between viruses, populations and the environment. Influenza A and B showed the most dynamic variability while RSV remained markedly consistent in this population. Influenza is zoonotic, constantly experiencing genetic shift and drift, and spilling over into human populations. However, humans are the reservoir for RSV, and RSV is therefore very well adapted to human populations and may not be as driven by geographic and population-level factors as influenza. For example, temperature and humidity patterns are known to influence the spread of both RSV and influenza. However, influenza epidemics are also influenced by sociodemographic characteristics of the underlying population (Chandrasekhar 2015, Yang et al. 2015). It is unclear exactly how much RSV is dependent on similar population features, but previous studies support associations between respiratory viruses and human behaviors that may help explain their seasonality (Sloan, Moore and Hartert, 2011).

Rhinovirus is also well-adapted to humans but shows far less regularity in its timing than RSV. In the Utah data, rhinovirus persists at comparatively low peak levels over a much longer period of time. This is in line with other studies, some of which suggest rhinovirus may be well-adapted to being present in populations at times other infections are not (Monto 2002). 

The delay of influenza season due to rhinovirus has been a subject of epidemiological interest for some time. For example, a particularly intense rhinovirus epidemic is hypothesized to have delayed 2009 H1N1 in France (Casalegno et al. 2010). Detecting interference at the population level may only be possible during more extreme events, such as pandemic influenza or higher than normal rhinovirus burden. Further studies with individual-level data and in different populations should be pursued to better understand why viral interference at the individual level does not necessarily translate to interference at the population level." 

2.) A limitation of this paper is that many others have examined the temporal relationship of viral epidemics, and these are not referenced here. Specifically, the interaction of influenza and RSV epidemics has been studied several times, with no clear pattern identified.

Response: Thank you. We agree that we needed to expand the reference list and have added several relevant articles to which we can compare our findings. The differences found between RSV and influenza at the individual level that have not been seen at the population level were the primary motivating factor for this study. 

3.) Another limitation is a description of why the statistical methods used here are preferable to other methods. Non-epidemiologists especially would need to know this, as the methods are not widely understood.

Response. We appreciate this comment. We added the below paragraph in section 2 to further explain our choice of methodology:

"Wavelet analysis and change point models are complementary methods we implemented to investigate the timing and patterns of epidemics. For our purposes, the strength of change point models over others such as SIR models or circular statistics is that they give more precise estimates of when different phases of a past epidemic began and terminated. Wavelet analysis compares epidemic timing in a way that allows for the behavior of the epidemic to change in different years. This would be more difficult if using methods that employ sine waves to estimate epidemic curves. We can therefore make direct comparisons of the historical characteristics of the epidemics (via change point models) and their synchrony with one another (wavelet analysis) using our selected methods. We recognize that there are many other methods that would lend different insights into the epidemic patterns described but deemed these two sufficient for the current study." 

4.) Perhaps the major limitation is the absence of a discussion of the meaning of the results. Where does this study advance our knowledge, and how do these data help us?

Response: We agree, and refer you to our response to comment #1, as we tried to address this comment as well in that response. 

Reviewer 3 Report

Callahan et al presented the comparative epidemiology of several seasonal respiratory viruses in Utah. The authors analysed the epidemic seasonality, synchrony and peak timing of each virus. They found that RSV dynamics are the most consistent of any of the viruses studied. Below are my comments:

1) Since the authors have the weekly case data of the respiratory viruses, it will be more preferable to estimate the peak time using circular statistics model what has higher precision than the change point model that has been used. 

2) It would be useful by citing and discussing about other comparative epidemiology studies e.g. J Infect doi:10.1016/j.jinf.2019.07.008. For example, the Introduction of this manuscript has pointed out that evidence of viral interference is also observed in other studies - e.g. RSV & PIV in J Infect doi:10.1016/j.jinf.2019.07.008.

3) Do the authors have access to the genetic sequence data of the respiratory viruses studied in the manuscript? The authors may discuss that state-of-the-art molecular epidemiology analyses of these genetic sequences could also provide insights into epidemic timing and transmission dynamics (Nature 2008 29;453(7195):615-9; PLoS Pathog 2014 10:e1003932.) in addition to what could be learn from incidence data. However, it might be useful to remind reader that the reference sequences of most of the non-influenza respiratory viruses are relatively insufficient which impedes such sequence analysis (Lancet Infect Dis 2017 17:e320-e326). 

Author Response

Dear Reviewer,

Thank you for your comments on the paper. They were very helpful and served to improve the paper. We respond to each comment below. 

1) Since the authors have the weekly case data of the respiratory viruses, it will be more preferable to estimate the peak time using circular statistics model what has higher precision than the change point model that has been used. 

Response: Thank you for this thoughtful idea. We carefully considered your suggestion of using circular statistics but decided to continue using change point analysis in conjunction with wavelet analysis. While circular statistics offer strengths in estimating peak and dispersion around the peak, we were interested in specifically comparing timing and duration of different phases of the seasonal epidemics. Change point models allowed us to estimate the specific dates that slopes changed along the curve. We now address this in the text in section 2 for clarification.

2) It would be useful by citing and discussing about other comparative epidemiology studies e.g. J Infect doi:10.1016/j.jinf.2019.07.008. For example, the Introduction of this manuscript has pointed out that evidence of viral interference is also observed in other studies - e.g. RSV & PIV in J Infect doi:10.1016/j.jinf.2019.07.008.

Response: Thank you. We agree the references needed to be expanded and treated more fully in the introduction and discussion. We added several references, including those you have listed. 

3) Do the authors have access to the genetic sequence data of the respiratory viruses studied in the manuscript? The authors may discuss that state-of-the-art molecular epidemiology analyses of these genetic sequences could also provide insights into epidemic timing and transmission dynamics (Nature 2008 29;453(7195):615-9; PLoS Pathog 2014 10:e1003932.) in addition to what could be learn from incidence data. However, it might be useful to remind reader that the reference sequences of most of the non-influenza respiratory viruses are relatively insufficient which impedes such sequence analysis (Lancet Infect Dis 2017 17:e320-e326). 

Response. Unfortunately we do not have access to any associated genetic data. We agree that genetic data would add useful insights, but we are not able to obtain it at this time. 

Round 2

Reviewer 2 Report

Minor point: Line 264-265 : "For example, temperature and humidity
patterns are known to influence the spread of both RSV and influenza"

While numerous studies indicate an association of temperature and humidity (and other meteorological features) with RSV activity, there is essentially no evidence that climate factors 'influence' RSV activity, as opposed to 'correlate with' RSV activity. The same is probably true for influenza as well.

Author Response

Reviewer:While numerous studies indicate an association of temperature and humidity (and other meteorological features) with RSV activity, there is essentially no evidence that climate factors 'influence' RSV activity, as opposed to 'correlate with' RSV activity. The same is probably true for influenza as well.

Response: Thank you for your careful review of the manuscript. While we agree that many studies focus on association or correlation without being able to prove causation, we disagree that evidence of causation does not exist. For example, there is evidence that changes in temperature (cold->warm) triggers the formation of the F-protein configuration needed to bind RSV to human cells in nasal passageways (Yunus et al. 2010). There is also the matter of how both influenza and RSV circulate according to different patterns in tropical vs. temperate climates. In 2008, Lowen et al. (2008) showed that transmission of influenza between guinea pigs was influenced by temperature and humidity. There are other studies along these same lines, but these should suffice for supporting the current point.